# A Comparison of the Safety, Efficacy, and Accuracy of Frame-Based versus Remebot Robot-Assisted Stereotactic Systems for Biopsy of Brainstem Tumors

**DOI:** 10.3390/brainsci13020362

**Published:** 2023-02-20

**Authors:** Chaoxi Li, Shiqiang Wu, Kuan Huang, Ran Li, Wei Jiang, Junwen Wang, Kai Shu, Ting Lei

**Affiliations:** Department of Neurosurgery, Tongji Hospital, Tongji Medical College, Huazhong University of Science and Technology, Wuhan 430034, China

**Keywords:** brainstem tumor, stereotactic surgery, biopsy, frameless, Remebot robot

## Abstract

Background: Brainstem tumors are rare and extremely heterogeneous and present significant challenges in surgical treatment. Thus, biopsies often set the foundation for the diagnosis of brainstem tumors. Multimodal, image-guided, robot-assisted frameless stereotactic biopsies are increasingly popular in neurosurgery centers. This study aimed to compare the safety, efficacy, and duration of the Remebot robot-assisted (Remebot) frameless brainstem tumor biopsy versus those of frame-based stereotactic biopsy. Method: A retrospective analysis of 33 patients with brainstem tumors who underwent stereotactic brainstem biopsies in the department of neurosurgery from January 2016 to January 2021 was conducted. The patients were divided into two groups: the Remebot group (*n* = 22) and the frame-based group (*n* = 11). The clinical characteristics, trajectory strategy, duration of procedure, diagnostic yielding, histopathological diagnosis, and postoperative complications were retrospectively analyzed and compared between the groups. Results: More pediatric patients performed Remebot frameless brainstem tumor biopsy than frame-based biopsy, with a mean age of 17.3 ± 18.7 vs. 32.8 ± 17.1 (*p* = 0.027). The diagnostic yield had no significant difference in the two groups, with the diagnostic yield of frame-based biopsy and Remebot frameless brain biopsy being 90.9% and 95.5%, respectively. The time of the total process was 124.5 min for the frame-based biopsy and 84.7 min for the Remebot frameless brain biopsy (*p* < 0.001). There were no significant differences with respect to the occurrence of complication or the duration of the operation between the two groups. Conclusion: Remebot frameless stereotactic brainstem biopsy is as safe and efficacious as frame-based stereotactic biopsy. However, Remebot frameless biopsy can reduce the total duration of the procedure and has better application in young pediatric patients. Remebot frameless stereotactic biopsies can be a better option towards the safe and efficient treatment of brainstem tumors.

## 1. Introduction

Brainstem tumors are a rare, highly heterogeneous, and challenging group of brain tumors with poor prognosis. They comprise 10% of all brain tumors in children and 1–2% in adults [1,2]. Efficient and specific treatment for these tumors remains challenging, as most of them are of a diffusely infiltrative nature and intolerant to craniectomy. A histological diagnosis is crucial to genetic, molecular studies and adjuvant treatment, required for further therapy including chemotherapy, radiotherapy, and potential targeted drug therapy. For a long time, the biological mechanism for these neoplasm remains obscure due to the historical association of surgery (even biopsy) with unacceptable morbidity and mortality. Following advances in surgical techniques and instruments, more and more studies have demonstrated that brainstem lesion biopsies can be conducted safely. Now, stereotactic brainstem biopsies have been designated an acceptable surgery and increasingly applied for these tumors, either conducted with frame-based or frameless systems.

Compared with standard frame-based stereotactic biopsies, robot-assisted frameless biopsies are highly efficient, safe, comfortable, and convenient. Many robot-assisted frameless stereotactic systems have been applied in neurosurgery, such as the Robotized Stereotactic Assistant (ROSA, Zimmer Biomet lnc., Warsaw, IN, USA); Neuromate Robot (Renishaw lnc., London, UK); NeuroArm Robot (Calgary, Alberta, Canada); and Remebot Robot (Baihuiweikang Technology Company, Beijing, China). The Remebot robot system has been widely applied in many neurosurgical centers in China [3,4]. The Remebot robot system comprises a planning system, a videometric tracking system, and an operating arm (Figure 1). The image data were collected into the planning system to build a three-dimensional image, define the surgical target, and design the trajectories. The videometric tracking system integrating three built-in cameras fixed on an independent stand was placed above the head, enabling the markers to be recorded for further registration. The markers on the patient’s head were recorded by the videometric tracking system and matched with the image in the planning system to accomplish the patient-to-robot registration. To validate the accuracy of the system, the operating arm moved toward the testing site to confirm the precision. According to the surgical protocols, finally, the operating arm moved to the planned targets to complete procedures including biopsy, suction, high temperature destruction, electrode implantation, and so on. This robotic system has been integrated in the surgeon’s armamentarium, trailblazing a new boulevard for diagnosis and treatment. Based on the fundamental experiences with the Remebot system, this study aimed to compare the safety, efficacy, and duration of Remebot frameless brainstem stereotactic biopsy with those of classical frame-based biopsy.

## 2. Methods

### 2.1. Patients

This retrospective study analyzed 33 consecutive patients benefitting from either Remebot frameless or frame-based stereotactic brainstem lesion biopsy at the Department of Neurosurgery, Tongji hospital, from January 2016 to January 2021. The included patients presented with a brainstem tumor without clear diagnosis, requiring a biopsy approved by a multidisciplinary neuro-radio-oncology board discussion. Brainstem tumors were defined by cerebral magnetic resonance imaging (MRI) as involving the mesencephalon, crus cerebri, pons, or medulla oblongata. The patients were divided into two groups: the frame-based group (Frame, *n* = 11) and the Remebot frameless group (Remebot, *n* = 22).

This study was approved by the ethics committee of Tongji Hospital, Huazhong University of Science and Technology. All patients and/or their relatives signed informed consent documents.

### 2.2. Surgical Procedure

All the surgeries were performed by the same neurosurgeon, Professor Kai Shu, Department of Neurosurgery, Tongji Hospital. Eleven cases received frame-based stereotactic brainstem tumor biopsy, and twenty-two underwent Remebot frameless brainstem biopsy.

For the standard frame-based stereotactic biopsy process, the Leksell Frame G stereotactic frame (Elekta lnc., Stockholm, Sweden) was placed on the patient’s head preoperatively under local anesthesia. Later, a preoperative MRI scan was performed, and then the target was carefully decided. The coordinates of the target were calculated and determined before the operation. Under general anesthesia, the surgeon made a burr hole with a diameter of 1 cm at the prescribed site and put a biopsy needle (Sedan-Vallicioni side-cutting needle (Elekta lnc., Stockholm, Sweden) with 2.5 mm diameter) into the brain lesion using either a supra (transfrontal/transtemporal) or an infratentorial (transcerebellar) approach. Finally, the wounds were sutured and disinfected after the specimens were collected with the standard suction–aspiration technique. All patients received a CT scan on the coming day.

For the Remebot frameless biopsy, the patients’ MRI data were obtained 1–2 days before surgery. To optimize the quality of imaging, 3.0 Tesla (GE HealthCare lnc, Waukesha, WI, USA) for a T1-weighted 3D gadolinium-enhanced thin-sliced (3D-bravo sequence) brain MRI and some special sequence (diffusion tensor imaging, magnetic resonance venography, etc.) was acquired. Six videometric marker stickers (Beijing Baihuiweikang Technology Company, Beijing, China) were properly attached to the temple and the top of the head without anesthesia 30 min before surgery for later laser-based surface registration (LSR). A full-head computed tomography (CT) scan (thickness: 0.625 mm) with markers was immediately performed for patients in a prone position (transcerebellar) or supine position (transtemporal). All the MRI and CT data were collected into the Remebot robot system and matched. Trajectory planning depended on tumor location and surrounding structures (cranial nerve, vascular and/or functional brainstem zones). The final trajectory was carefully designed based on the entry point and biopsy target, with either a transcerebellar or transtemporal approach. The patient’s head was immobilized by a Mayfield clamp(Interra neurosciences lnc,. Plainsboro, NJ, USA) under general anesthesia in the prone position (transcerebellar) or supine position (transtemporal). After accurate laser-based surface registration with four scalp markers and verification with two test markers, a burr hole was drilled at the entry point, and biopsy specimens were collected with the suction–aspiration technique and sent to pathology. Finally, the wound was disinfected and sutured. All patients were reexamined by CT on the coming day.

### 2.3. Data Acquisition

All patient files were recovered from the Department of Medical Records, including demographics, imaging information, medical documentation, histopathological diagnosis, treatment, and follow-up. The length of trajectory was determined by the distance from the entry point to the target. The duration of process represents the efficiency of the surgery, including registration and operation time. The diagnostic yielding represents the accuracy of the biopsy. Low complications represent the safety of surgery.

### 2.4. Statistical Analysis

Statistical analysis was conducted with SPSS Statistics 22.0 (IBM Corporation, Armonk, NY, USA). Continuous data are described as $\bar{x}\pm s$. The intergroup comparison of the categorical variable was analyzed using the Chi-Squared test or Fisher’s exact test (when case number of any group < 5). Continuous variables were compared with Student’s *t*-test. *p* < 0.05 was considered to be a statistically significant difference.

## 3. Results

We included 22 patients who were treated with robot-assisted frameless stereotactic brainstem biopsy and 11 who were treated with frame-based stereotactic brainstem biopsy in a 5-year period (January 2016–January 2021). All the demographic and detailed information of patients is listed in Appendix A.

### 3.1. Clinical Characteristics

Clinical characteristics of the cohort are illustrated in Table 1. The mean age at biopsy was 22.0 ± 19.3 years, with biopsy conducted on 18 children (<16 years old, 54.5%) and 15 adults (45.5%). There were more pediatric patients that underwent the Remebot robot-assisted brainstem tumor biopsy than frame-based biopsy, with a mean age of 17.3 ± 18.7 versus 32.8 ± 17.1 (*p* = 0.027). The sex ratio was 15:18 (male:female). The most common symptoms at diagnosis were gait impairment (ataxia); vertigo; motor deficit; facioplegia; and increased intracranial pressure (headache, nausea, and vomiting). The most common region with lesions was the pontine region (31/33, 93.9%), followed by the mesencephalon (23/33, 69.7%). No biopsy was performed for lesions in the medulla oblongata. There were no significant differences in sex ratio, symptoms, or localization of the lesion between the Remebot group and frame-based group.

### 3.2. Procedure and Complications 

Details of the procedure and complications of the cohort are illustrated in Table 2. Most biopsies were conducted with the infratentorial (transcerebellar) approach (25/33, 75.8%) (Figure 2). In eight patients, the biopsy was performed with the supratentorial approach (transfrontal or transtemporal) (8/44, 24.2%) (Figure 3). More patients in the frame-based group underwent midbrain biopsy than those in the Remebot group (6/11 vs. 2/20, *p* = 0.008). The infratentorial approach for biopsy was used more often in the Remebot group than in the frame-based group (20/22 vs. 6/11, *p* = 0.032). Therefore, the trajectory length of biopsy in the frame-based group was longer than that in the Remebot group (91.11 ± 5.71 mm vs. 71.27 ± 1.49 mm, *p* < 0.001). There were no significant differences in the operation time between the Remebot and frame-based groups. However, the overall procedure duration was shorter in the Remebot group than the frame-based group. There was no postoperative hemorrhage needing a second operation in either group. Two patients developed transient complications including diplopia and nystagmus in each group. All patients’ symptoms were completely resolved within 3 months. There were no permanent deficits or deaths related to the biopsy procedures. There was no statistically significant difference in the occurrence of complication between the two groups.

### 3.3. Histopathology

The Histopathologic finding of the cohort are illustrated in Table 3. Histopathologic diagnosis was achieved in 31 cases (31/33, 93.9%). There was no significant difference in diagnostic yielding between the Remebot group (21/22, 95.9%) and the frame-based group (10/11, 90.9%) (*p* > 0.999). Each group reported one nonconclusive diagnosis, possibly because of the amount and quality of the specimen. The three most common tumor entities encountered were diffuse low-grade glioma grade II (48.5%, 16/33); high-grade glioma grade IV (36.4%, 12/33); and lymphoma (2/33, 6.1%). There was no statistical difference in low-grade glioma or high-grade glioma frequency between the Remebot and frame-based groups. Notably, seven cases of histone 3(H3) K27M-mutant diffuse midline gliomas were noted in the diffuse high-grade glioma group.

### 3.4. Treatment and Follow-Up

Based on the histopathologic findings, most cases received postoperative treatment including radiotherapy (31/33, 93.9%) and chemo-radiotherapy (14/33, 42.4%). Patients without a conclusive diagnosis were closely followed up. The mean follow-up duration was 25.4 months, and the median overall survival was 18.5 months (range: 3–69 months). At the time of writing this report, 17 patients had died and 16 were alive.

## 4. Discussion

In 2021, the WHO updated an integrated molecular and histological diagnostic framework for CNS tumors [5]. The updated molecular classifications have huge implications for prognosis, therapeutic options, and potential clinical trial accessibility. The need for accurate molecular diagnosis drives the demand for the acquisition of diagnostic tissue, especially for anatomically unresectable brainstem tumors. Stereotactic brain biopsy has been widely applied as an accurate, effective, and safe procedure for the diagnosis of brain lesions [6,7,8,9]. With development in artificial intelligence and neuronavigation systems, robot-assisted frameless neurosurgery has been widely performed and has expanded the depth of stereotactic and functional neurosurgery. The Remebot robot is a robot-assisted stereotactic system produced in China and has been successfully applied in varied clinical scenarios. We have tested the accuracy of the Remebot system in many previous applications and have verified its accuracy and safety. This study illustrates the experience with a series of brainstem tumor biopsies using frame-based and Remebot robot-assisted frameless systems under similar circumstances, that is, at a single center, by the same surgeon, and under the guidance of a single surgical team.

It is well known that the fundamental aim of stereotactic brain biopsy is to maximize the accuracy of the biopsy and minimize complications. To the best of our knowledge, a high diagnostic yield resulting from high biopsy accuracy and low complications benefit from microsurgical techniques and short procedure duration. The current robot-assisted frameless systems are capable of achieving an accuracy of <3 mm and can provide a comparable degree of spatial accuracy and consistency to that of frame-based systems [10,11,12]. Furthermore, although the robot-assisted frameless systems have similar operation times, they have a shorter duration of preoperative procedure than the traditional frame-based systems. Nevertheless, the robot system is space-occupying and cost-consuming, which hampers its application in less developed hospitals. Moreover, the whole procedure necessitates good training of the neurosurgeon, since a system bug could be anywhere and anytime at the beginning. Therefore, it is quite common that the robotic engineers participate in the surgical plan and assist the operation of newly installed robot systems.

In the study, 33 brainstem lesion biopsies were conducted with the Remebot robot (*n* = 22) and frame-based (*n* = 11) techniques. The mean age of the patients at biopsy in the Remebot group was significantly younger than that in the frame-based group, mostly owing to the inclusion of more young pediatric cases (<9 years) who were uncooperative in the Remebot group. Additionally, more patients underwent the supratentorial approach targeting the midbrain in the frame-based group than in the Remebot group, because the supratentorial trajectory in the frame-based group has the advantage of easier patient positioning and frame adaptations [13]. For most cases involving the pontine region, the infratentorial approach (transcerebellar) was used. Several regions were reportedly chosen to resect lesions located in the pons, which included the supratrigeminal zone, peritrigeminal zone, lateral pontine zone, median sulcus, supracollicular zone, and infracollicular zone [14]. However, there has been little study focus on the safety of the region selection in brainstem stereotactic biopsy, since biopsying with a needle is much less invasive than performing a surgical resection. We usually target the tumor located in the lateral pontine zone without severe complications.

In terms of the total duration of the biopsy, the duration of the frame-based biopsy was significantly longer than that in the Remebot group (124.5 ± 2.78 vs. 84.73 ± 2.19 min, *p* < 0.001), which is in good accordance with the previous study [3]. Nevertheless, there was no statistically significant difference in the surgical time (44.14 ± 1.40 vs. 45.45 ± 2.67 min, *p* = 0.632) between the two groups. This was likely because more time was spent on the preoperative preparation in the frame-based group (i.e., local anesthesia, frame adapted with pins), whereas the marker just needed to be affixed on the head of patient in the Remebot group, which was convenient and time saving.

The overall diagnostic yield was 93.9%, with 95.5% in the Remebot group and 90.9% in the frame-based group, respectively (*p* > 0.05). As reported by previous research, frameless stereotactic biopsy can obtain a diagnostic yield from 89% to 99.3%, and a frame-based procedure can provide one of 81.3–99.2% [15,16,17,18,19,20]. There were two cases (2/22, 9.1%) that presented transient new deficits postoperatively in the Remebot group, and they completely recovered within 3 months. Two patients (2/11, 18.2%) in the frame-based group experienced postoperative deficit, but recovered in 3 months. There was no statistically significant difference in the occurrence of complications between the two groups. It seems like more complications result from the supratentorial approach (transtemporal and transfrontal), though without a significant difference (*p* = 0.07). These data are consistent with the diagnostic yield (96.1%), morbidity (6.7%), and mortality (0.6%) reported in a meta-analysis of 735 stereotactic biopsies of pediatric brainstem tumors [21].

There is still an ongoing debate regarding the accuracy between laser-based surface registration (LSR) and bone fiducial registration (BFR) [22,23]. LSR is more convenient and acceptable for patients (especially pediatric cases), while studies have shown relatively higher inaccuracy compared to BFR (especially in long trajectories). All patients in the Remebot group had biopsies conducted with the laser-based surface registration (LSR) and obtained satisfactory accuracy. From our experience, we performed four markers for registration and two for verification, and then one anatomic marker for verification (mostly the lambda) to guarantee accuracy.

In a conventional frame-based stereotactic biopsy, brainstem tumors have been sampled more frequently by the transfrontal approach (TFA) because of the requirement of a prone position and specific attachment of the stereotactic ring [19]. The most widely applied robot-assisted stereotactic trajectory is the transcerebellar approach (TCA), which provides a comparatively wider stereotactic corridor and shorter trajectory length [7]. Most of our biopsies were conducted with the TCA (75.8%, 25/33), except for six patients in whom midbrain biopsy was performed using the transtemporal approach (TTA). There are only few studies that have applied TTA for brainstem biopsy [6], potentially because of a more vascular structure in the corridor and less accessibility in certain directions of the brainstem lesion. We used TTA to perform brainstem lesion biopsy in six selected patients and obtained diagnostic success without surgical morbidity. To avoid the difference in the skull size between adult and pediatric patients, we only compared the trajectory length between TCA and TTA in pediatric cases. TTA has a relatively longer trajectory length than TCA (90.7 ± 5.5 vs. 71.4 ± 3.3 mm, *p* < 0.001), but much shorter than TFA (>12 cm) [7]. To minimize the length of the trajectory, TTA is still a good option for some selected patients.

The majority of our cohort comprised diffuse brainstem tumors (75.8%, 27/33), which can hardly be removed by surgery and highly depend on histological diagnosis for prognosis and further treatment. There were 16 diffuse low-grade gliomas (most are diffuse astrocytomas) and 11 diffuse high-grade gliomas (most harbored the H3K27M mutation) among our patients, which were predominantly diagnosed in pediatric patients. Diffuse intrinsic pontine gliomas (DIPGs) are a universally fatal brainstem tumor and are not suitable for a biopsy for many years [20,24]. The subsequent lack of eligible pathologic tissue hindered both the exploration of biologic mechanisms and the development of preclinical models. Within the past decade, the renaissance in brainstem biopsies has provided a scaffolding for the investigations of DIPG to build on [20]. Subsequently, the H3K27M mutation was discovered to increase epigenetically driven transcriptional expression in DIPG [25,26]. Expression of H3K27M causes a decrease in H3K27 methylation and an increase in histone acetylation. Histone deacetylase inhibitor (HDACi) has shown efficacy in several preclinical diffuse midline glioma models and has potential clinical applications [24,27]. Recently, GD2-CART cell therapy of H3K27M-mutated diffuse midline gliomas showed promising clinical benefit [28].

The inclination for a diffuse glioma to occur in the brainstem precludes resection and even biopsy. The stereotactic biopsy has only been performed to be safe in selected patient in recent years. There is still much less biopsy performed for lesions in the medulla oblongata, in which there exist significant complication and surgical risks that cannot be ignored. Even when samples can be collected stereotactically, samples are not always sufficient for histological and genetic diagnosis, and a repeat biopsy is rarely advised. Therefore, there is an urgent need to provide patients with brainstem tumors a safe alternative to a stereotactic biopsy that can be reliable and repeatedly collected without significant risks. Such an alternative method has garnered great interest is that of biological fluids, which include cerebral spinal fluid (CSF), plasma, urine, and saliva. It has been demonstrated that tumor nuclear material associated with the presence of solid tumors can be detected in a patient’s biological fluid (liquid biopsy). These materials are collectively referred to as circulating tumor DNA (ctDNA). The most commonly studied form of ctDNA is the circulating tumor cell, which sheds from the tumor entity and accesses a biological fluid system. Another form of ctDNA is the exosome, an extracellular vesicle released from the membrane system of tumor cells that carries genetic characteristics of the tumor [29]. Tissue biomarkers are also an excellent source of noninvasive monitoring, which can help monitor treatment response and provide important prognostic value. A great deal of work is being conducted to identify biomarkers in DIPG. The tumor-specific proteins cyclosporin A and dimethylarginase 1 have been isolated in the cerebrospinal fluid, which were expressed at low levels in serum and urine.

There were several limitations in the study. Firstly, this is a retrospective study with data collection limited within a 5-year period since the Remebot system was applied. Secondly, the size of the cohort was still not large enough, which limits the power of statistical analysis. This is mostly owing to the low occurrence of brainstem tumors, especially those requiring a biopsy approved by the multidisciplinary neuro-radio-oncology board. Thirdly, improving diagnostic yield should be discussed and a perspective study should be carefully designed and performed. Lastly, the study lacks detailed postoperative treatment information during the follow-up. Since the patients were assigned to different departments (even in different hospitals) at certain stages, the relevant data were not always available. It would be of great significance to explore the following treatments after biopsy and the survival benefits. 

## 5. Conclusions

The Remebot frameless biopsy procedure was verified to be comparatively as effective and safe as the frame-based stereotactic procedure for brainstem tumor biopsy. Furthermore, the Remebot robot system has wider applications in young pediatric patients and has a short total procedure duration. Frameless robot-assisted stereotactic biopsies can offer a better platform towards the safe and efficient treatment of brainstem tumors.

## Figures and Tables

**Figure 1 brainsci-13-00362-f001:**
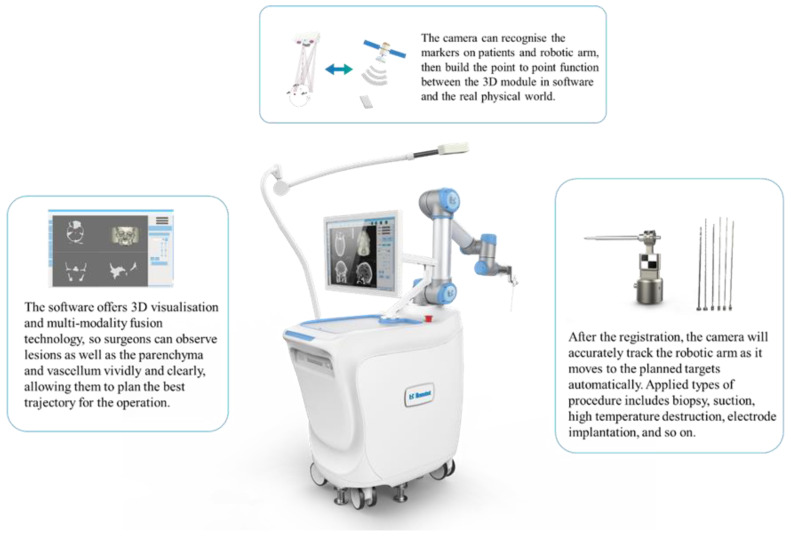
Image of the Remebot system.

**Figure 2 brainsci-13-00362-f002:**
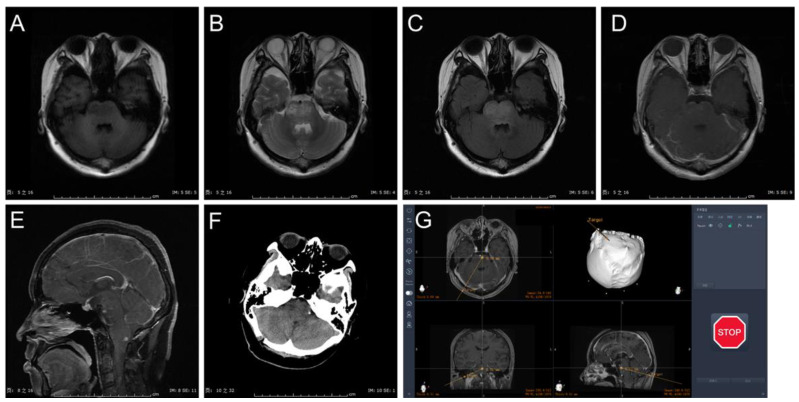
Case presentation: (**A**) MRI with T1 sequence in axial view. (**B**) MRI T2 sequence in axial view. (**C**) MRI T2 flair sequence in axial view. (**D**) MRI T1 gadolinium sequence in axial view. (**E**) MRI T1 gadolinium sequence in sagittal view. (**F**) 1-day postoperative CT scan. (**G**) Surgical plan with transcerebellar approach in Remebot system.

**Figure 3 brainsci-13-00362-f003:**
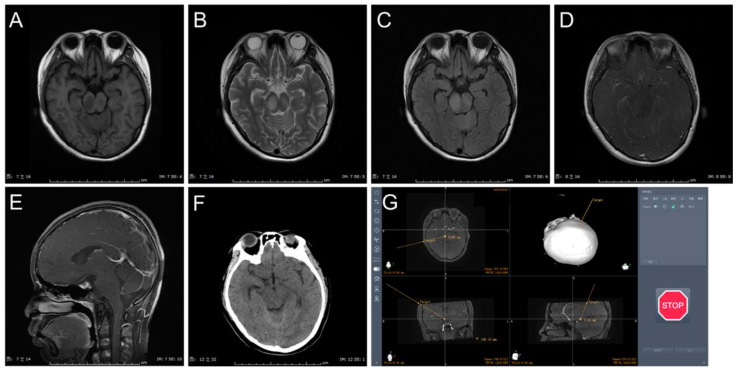
Case presentation: (**A**) MRI with T1 sequence in axial view. (**B**) MRI T2 sequence in axial view. (**C**) MRI T2 flair sequence in axial view. (**D**) MRI T1 gadolinium sequence in axial view. (**E**) MRI T1 gadolinium sequence in sagittal view. (**F**) 1-day postoperative CT scan. (**G**) Surgical plan with transtemporal approach in Remebot system.

**Table 1 brainsci-13-00362-t001:** Summary of the patients’ clinical characteristics.

	Remebot Robot Group (*n* = 22)	Frame-Based Group (*n* = 11)	*p*
Age (mean ± SD), years	17.3 ± 18.7	32.8 ± 17.1	0.027
Sex ratio (male/female)	7:15	7:4	0.136
Symptoms			
Vertigo	5	4	0.438
Ataxia	8	5	0.714
Motor deficit and/or sensory deficit	5	4	0.438
IIP	7	4	>0.999
Region of biopsy			
Midbrain	2	6	0.008
Pons	20	5	0.008

IIP: Increased intracranial pressure (headache and vomiting).

**Table 2 brainsci-13-00362-t002:** Comparison of the trajectory, trajectory length, total procedure duration, operation time, and complication.

	Remebot Robot Group (*n* = 22)	Frame-Based Group (*n* = 11)	*p*
Trajectory (supra: infratentorial)	2/20	6/5	0.032
Trajectory length, mm	71.27 ± 1.49	91.11 ± 5.71	<0.001
Total procedure duration, mean, min	84.73 ± 2.19	124.5 ± 2.78	<0.001
Operation time, mean, min	44.14 ± 1.40	45.45 ± 2.67	0.632
Complication	2/20	2:9	0.586

**Table 3 brainsci-13-00362-t003:** Histopathological diagnosis of the two groups.

Histopathological Finding	Remebot Robot Group (*n* = 22)	Frame-Based Group (*n* = 11)	*p*
Diagnostic yield	21/22	10/11	>0.999
Diffuse low-grade glioma	11	5	
Diffuse high-grade glioma	8	4	
Diffuse large B cell lymphoma	1	1	
Nonconclusive diagnosis	1	1	

## Data Availability

Not applicable.

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
