# Peer review of "A Comparison of the Safety, Efficacy, and Accuracy of Frame-Based versus Remebot Robot-Assisted Stereotactic Systems for Biopsy of Brainstem Tumors"

_brainsci, 2023, doi:10.3390/brainsci13020362_

Round 1

Reviewer 1 Report

The authors report their experience on stereotactic needle biopsy of brainstem tumors comparing retrospectively a series frame based (11 patients) with one in which a frame-less robot (Remebot) assisted technique was used (22 patients).

The results showed that the diagnostic yield was high in both series (> 90%) and the complications’ rate was low in both, with less percentage in the Remebot group. The length of surgery was shorter in the robot-assisted series that takes advantage from the laser-based registration, without a significant difference from the statistical point of view. The authors’ conclusions are that the robot-assisted stereotactic biopsy of brainstem tumors is as safe and efficacious as the frame-based one.

Considerations.

Prognosis of brainstem gliomas is still demeaning, but identifying specific molecular features would help in understanding better the oncological mechanisms and would also furnish the possibility of a targeted therapy, as correctly reported by the authors. Planning the safer trajectory to the tumor core and reproducing it during live surgery are the crucial issues of this chapter.

The present paper confirms what has been the experience of the last decade: the frameless stereotactic surgery is as reliable as the frame-based one and the robot-assisted technique might add accuracy and support to the surgeon. The routes (trans-frontal, trans-temporal or trans-cerebellar) may be different accordingly to the surgeons’ preference and experience, but each one with the right trajectory may be safely utilized.

Unfortunately, there are still many oncological centers performing radiotherapy of brainstem gliomas without histology, and that should be avoided. 

Author Response

   We thanks to the reviewer’s positive comments,and hope our work will be helpful to the academic community.

Reviewer 2 Report

The authors compared the efficacy, safety, and accuracy of frame-based versus Remebot robot-assisted frameless stereotactic systems for biopsy of brainstem tumors. The study is interesting, however, I have some concerns to discuss.

-This cohort is too small.

-Please state the disadvantages of this method.

-There was no statistically significant difference between the two groups with regard to complication rates or operative time, but that does not make this method superior. 

Author Response

We thanks to the reviewer’s pertinent comments, and replied as follow.

  1. One of the limitations of the study is the size of the cohort was still not large enough, which limits the power of statistical analysis. It is mostly owing to the low occurrence of brainstem tumor, especially those requiring biopsy and approved by the multidisciplinary neuro-radio-oncology board.
  2. The major disadvantage of the Remebot robot-assisted stereotactic systems is that : the robot system is space-occupying and cost-consuming, which hamper its application in less developed hospital. Moreover, the whole procedure necessitate well-training of the neurosurgeon . Therefore ,it’s quite common that the robotic engineers participated the surgic plan and assisted the operation for newly installed robot system.
  3. There was no statistically significant difference between the two groups with regard to complication rates or operative time. However, Remebot frameless biopsy can reduce the total duration of the procedure and has better application in young pediatric patients, which make it superior in certain circumstance.

Reviewer 3 Report

 Brainstem tumors are rare and extremely heterogeneous and present significant challenges in surgical treatment. Thus, biopsies often set the foundation for the diagnosis of brainstem tumors. Multimodal, image-guided, robot-assisted frameless stereotactic biopsies are increasing popular in neurosurgery centers.

Authors  aimed to compare the efficacy, safety, and duration of the Remebot robot-assisted frameless brainstem tumor biopsy versus those of frame-based stereotactic biopsy.

 They performed a retrospective analysis of 33 patients with brainstem tumors who underwent stereotactic brainstem biopsies in their department from January 2016 to January 2021 was performed. We divided the patients into two groups: the frame-based group (n=11) and the Remebot robot group (n=22).

The clinical characteristics, trajectory strategy, duration of procedure, diagnostic yielding, histopathological diagnosis, and postoperative complications were retrospectively reviewed and compared between these two groups.

Their Results showed that:

(a) more pediatric patients underwent Remebot robot-assisted brainstem tumor biopsy than frame-based biopsy, with a mean age of 17.3±18.7 vs. 32.8±17.1 (p=0.027).

 (b) No significant difference in diagnostic yield was detected in the two groups, with the diagnostic yield of frame-based biopsy and Remebot robot-assisted frameless brain biopsy being 90.9% and 95.5%, respectively.

(c)The duration of the total procedure was 124.5 min for the frame-based biopsy and 84.7 min for the Remebot robot-assisted frameless brain biopsy (p<0.001).

(d)  There were no statistically significant differences with respect to complication rate or operation time between the two groups.

They concluded stating that: (1) Remebot robot-assisted stereotactic brainstem biopsy is as safe and efficacious as stereotactic frame-based biopsy.(2) However, frameless biopsy can reduce the total duration of the procedure and has better application in young pediatric patients. (3)Frameless robot-assisted stereotactic biopsies can provide a better platform towards safe and efficient management for brainstem lesions.

The study is interesting.

I have some minor suggestions:

1)      Introduction must be enlarged introducing more background.

2)      Introduce the description of figure 1.

3)      The statistics “Data are described as ?̅± ?. The intergroup comparison was performed using Student’s t-test and χ2 test or Fisher’s exact test (case number: <5). P<0.05 was considered to indicate statistically significant differences.” Is described criptically. Please enlarge and use more care in the description of the methodology.

4)      Avoid the use of “we” and “our”.

5)      Avoid the use of small paragraphs. (see for example par. 3.4 , 3.5).

6)      Insert the limitations of the study in the discussion.

7)      Avoid the use of bold in the body of the  manuscript.

Author Response

We thanks to the reviewer’s comprehensive comments, and replied as follow.

  1. Introduction was not enough in the first manuscript we expanded the introduction with more background.
  2. To make it more clear to the reader, we add more description of figure 1 in the introduction.
  3. We clarify the statistical method to make it mor clear.
  4. We avoided the use of  “we”  and  “our”.
  5. We avoided the use of small paragraph and combined the par 3.4 and 3.5.
  6. We added the limitation of the study in the discussion.
  7. We avoided the use of bold font in the manuscript as the journal required.

Round 2

Reviewer 2 Report

The authors replied well, so the manuscript is suitable for publication.